# DNA methylation analyses identify an intronic *ZDHHC6* locus associated with time to recurrent stroke in the Vitamin Intervention for Stroke Prevention (VISP) clinical trial

Nicole M. Davis Armstrong[1], Wei-Min Chen[2,3], Fang-Chi Hsu[4], Michael S. Brewer[1], Natalia Cullell[5,6], Israel Fernández-Cadenas[5,6], Stephen R. Williams[7], Michèle M. Sale[2,3], Bradford B. Worrall[3,7], Keith L. Keene[1,8]*

**1** Department of Biology, East Carolina University, Greenville, NC, United States of America, **2** Center for Public Health Genomics, University of Virginia, Charlottesville, VA, United States of America, **3** Department of Public Health Sciences, University of Virginia, Charlottesville, VA, United States of America, **4** Department of Biostatistics and Data Science, Division of Public Health Sciences, Wake Forest School of Medicine, Winston-Salem, NC, United States of America, **5** Stroke Pharmacogenomics and Genetics, Fundació Docència i Recerca Mútua Terrassa, Hospital Universitari Mútua de Terrassa, Terrassa, Barcelona, Spain, **6** Stroke Pharmacogenomics and Genetics, Sant Pau Institute of Research, Hospital de la Santa Creu i Sant Pau, Barcelona, Spain, **7** Department of Neurology, University of Virginia, Charlottesville, VA, United States of America, **8** Center for Health Disparities, Brody School of Medicine, East Carolina University, Greenville, NC, United States of America

\* keenek@ecu.edu

**Data Availability Statement:** Individual level genetics and epigenetics data is considered sensitive controlled data and cannot be shared

## Abstract

Aberrant DNA methylation profiles have been implicated in numerous cardiovascular diseases; however, few studies have investigated how these epigenetic modifications contribute to stroke recurrence. The aim of this study was to identify methylation loci associated with the time to recurrent cerebro- and cardiovascular events in individuals of European and African descent. DNA methylation profiles were generated for 180 individuals from the Vitamin Intervention for Stroke Prevention clinical trial using Illumina HumanMethylation 450K BeadChip microarrays, resulting in beta values for 470,871 autosomal CpG sites. Ethnicity-stratified survival analyses were performed using Cox Proportional Hazards regression models for associations between each methylation locus and the time to recurrent stroke or composite vascular event. Results were validated in the Vall d'Hebron University Hospital cohort from Barcelona, Spain. Network analyses of the methylation loci were generated using weighted gene coexpression network analysis. Primary analysis identified four significant loci, cg04059318, ch.2.81927627R, cg03584380, and cg24875416, associated with time to recurrent stroke. Secondary analysis identified three loci, cg00076998, cg16758041, and cg02365967, associated with time to composite vascular endpoint. Locus cg03584380, which is located in an intron of *ZDHHC6*, was replicated in the Vall d'Hebron University Hospital cohort. The results from this study implicate the degree of methylation at cg03584380 is associated with the time of recurrence for stroke or composite vascular events across two ethnically diverse groups. Furthermore, modules of loci were associated with clinical traits and blood biomarkers including previous number of strokes, prothrombin fragments 1 + 2, thrombomodulin, thrombin-antithrombin complex, triglyceride levels, and tissue

publicly, as specified by the IRBs of Wake Forest University School of Medicine, the University of North Carolina at Chapel Hill School of Medicine, and the University of Virginia School of Medicine, along with the NIH Data Access Committee. Controlled-access data can only be obtained if a user has been authorized by the appropriate Data Access Committee. The individual level Genomics and Randomized Trials Network (GARNET) VISP data are available in the database of Genotypes and Phenotypes (dbGaP) (Accession: phs000343.v3.p1) and can be requested through the dbGaP Authorized Access System (https://dbgap.ncbi.nlm.nih.gov/aa/wga.cgi?page=login). The authors will also share the data on request.

**Funding:** This work was supported by the National Institutes of Health, grant U01HG005160 and Supplement U01HG005160-03S1 (MMS/BBW) from the National Human Genome Research Institute, an American Heart Association Scientist Development Award 12SDG9180012 (KLK), and funds from the Walking for My Life 5K Stroke Walk sponsored by Faith Christian Center International (Charlottesville, VA). The original study recruitment and datasets for the VISP clinical trial were supported by research grant R01 NS34447 (JT) from the National Institute of Neurological Disorders and Stroke. The funders had no role in study design, data collection and analysis, decision to publish, or preparation of the manuscript.

**Competing interests:** The authors have declared that no competing interests exist.

plasminogen activator. Ultimately, these loci could serve as potential epigenetic biomarkers that could identify at-risk individuals in recurrence-prone populations.

## Introduction

Ischemic strokes account for 87% of all strokes and are heterogeneous, multifactorial diseases comprising genetic and environmental contributions. Of the approximately 795,000 incident stroke cases annually in the United States, approximately 25% are recurrent events [1]. Individuals experiencing an ischemic stroke are at high risk of having a recurrent stroke or developing cardiovascular diseases including myocardial infarction (MI), coronary heart disease, or vascular death. Only a proportion of the estimated 37.9% heritability for ischemic stroke [2] is accounted for from genetic variants identified in genome-wide association studies, suggesting other mechanisms, such as epigenetic modifications, could comprise some of the remaining heritability for ischemic stroke and stroke recurrence risk.

Epigenetics, such as DNA methylation, refer to chemical modifications of DNA structure that can be maintained over cellular generations [3] and serve to propagate cellular memory [4]. Abnormal DNA methylation patterns have been implicated in a number of cardiovascular diseases [5, 6]; however, few studies have investigated these epigenetic contributions to stroke recurrence [7–9]. The aim of this study was to elucidate DNA methylation loci associated with the time to vascular events, including recurrent stroke, MI, and death. Single locus and comprehensive loci networks were analyzed in 180 individuals from the Vitamin Intervention for Stroke Prevention (VISP) clinical trial and further validated in the independent Vall d'Hebron University Hospital cohort. Findings from this study support the utility of epigenetic marks as potential biomarkers and may lead to improved prognosis or prevention of recurrent stroke and cardiovascular events.

## Methods

### Ethics statement

The institutional review boards (IRBs) of Wake Forest University School of Medicine, the University of North Carolina at Chapel Hill School of Medicine, and individual recruitment sites approved the VISP clinical trial study protocol. All VISP participants provided written, informed consent. A subset of 2,100 participants agreed to be included in subsequent genetic studies. IRB approval from the University of Virginia and East Carolina University was obtained for the genetic and epigenetic components.

### Discovery cohort: Vitamin Intervention for Stroke Prevention (VISP) clinical trial

VISP was a multi-centered, double-blinded, randomized and controlled clinical trial that enrolled participants aged 35 years or older with baseline homocysteine levels at or above the 25th percentile and was designed to determine whether pyridoxine (B6), cyanocobalamin (B12), and folic acid (B9) supplementation reduced recurrent cerebral infarction, MI, or fatal coronary heart disease (CHD) [10]. Participants were enrolled within 120 days of suffering a non-disabling cerebral infarction, assigned a daily B vitamin high-dose or low-dose formulation, and followed for two years. While ischemic subtype was not adjudicated, based on inclusion/exclusion criteria [10], the VISP enrollment stroke was most likely a small-vessel (lacunar) infarct.

A subset of 2,100 VISP participants consented for inclusion in genetic studies of which, methylation data were generated for 204 participants. Upon quality control (QC), the methylation profiles of 180 individuals were used in subsequent analyses, including 76 individuals of African descent (AFR) and 104 of European descent (EUR). Individual level genetics and epigenetics data is considered sensitive controlled data and cannot be shared publicly, as specified by the IRBs of Wake Forest University School of Medicine, the University of North Carolina at Chapel Hill School of Medicine, and the University of Virginia School of Medicine, along with the NIH Data Access Committee. Controlled-access data can only be obtained if a user has been authorized by the appropriate Data Access Committee. The individual level Genomics and Randomized Trials Network (GARNET) VISP data are available in the database of Genotypes and Phenotypes (dbGaP) (Accession: phs000343.v3.p1) and can be requested through the dbGaP Authorized Access System (https://dbgap.ncbi.nlm.nih.gov/aa/wga.cgi?page=login). The authors will also share the data on request.

## Replication cohort: The Vall d'Hebron University Hospital cohort

From a cohort of 1,900 patients with stroke from Vall d'Hebron University Hospital (Barcelona, Spain), 28 subjects with composite vascular recurrence events were selected. Of these participants, 18 had a recurrent ischemic stroke. Composite vascular recurrence was described as new ischemic stroke, MI, peripheral vascular disease, or cardiovascular death and was detected through telephone interviews every three months or direct clinical visit. The epigenome-wide methylation profiles were generated using Illumina Infinium 450k BeadChip microarrays. The profiles were processed in a single working batch and preprocessing, correction, normalization steps, and QC was performed using R, as previously described [8]. The epigenetics and genetics summary data used for replication will be shared for research studies on approval of the principal investigators of the GRECOS (Genotyping Recurrence Risk of Stroke) cohort.

## Stroke recurrence and composite vascular event definition

The primary vascular endpoint analyzed in the current study was an incident recurrent stroke during the VISP trial (VISP recurrent stroke). VISP recurrent stroke was defined as an acute neurological ischemic event, requiring a sudden onset of symptoms lasting at least 24 hours, as reported on the Follow-up Stroke Symptoms (FSS) form or as determined by the VISP Endpoint Review Committee [10].

A composite vascular event was examined as a secondary endpoint and was defined as fatal coronary heart disease, a non-fatal hospitalized MI and resuscitation for cardiac collapse, coronary bypass surgery, coronary angioplasty, or a VISP recurrent cerebrovascular event.

## Methylation data generation

Genomic DNA was extracted from whole blood samples of the VISP participants, as previously described [9]. Briefly, the DNA was denatured and bisulfite-converted using Zymo EZ DNA Methylation Kits (Zymo Research Corp., Irvine, CA). Illumina Infinium Human Methylation450k BeadChip microarrays were used to interrogate CpG sites across the genome and the resulting intensity files were analyzed using GenomeStudio. Beta scores were generated as the ratio of methylated intensities divided by the sum of the methylated and unmethylated intensities (i.e. the proportion of total signal from the methylation-specific probe). Probes were removed if they failed to hybridize (detection $p > 5\%$) or if they were located on sex chromosomes. The filtered beta scores underwent stratified quantile normalization with the *minfi* package in R [11–13]. A total of 470,871 autosomal CpGs were used in subsequent analyses.

## Covariate generation

To account for population substructure, a principal component analysis was performed using KING software [14] on VISP genotype data. The top four and ten principal components (PCs) were generated and used as covariates in the survival analysis for the EUR population and AFR population, respectively. Batch effect was adjusted for using factor variables indicative of methylation data generation round, while cellular heterogeneity due to the variation in cell population proportions within whole blood samples was controlled for using cell proportion estimates generated by the estimateCellCounts function in *minfi* [12, 15–17]. This function generates cell counts for B-lymphocytes, CD4+ and CD8+ T-lymphocytes, natural killer cells, granulocytes, and monocytes.

## Statistical analysis

Baseline characteristics of the study participants with and without recurrent stroke were compared using t-tests and $\chi^2$ tests for continuous and categorical traits, respectively, by each ethnic stratum.

Survival analyses utilizing Cox Proportional Hazards (PH) regression models were performed for the time in days to VISP recurrent stroke or composite vascular event for both AFR and EUR participants independently. The degree of methylation was the exposure and the models adjusted for age, sex, PCs, batch effect, treatment arm, and cellular proportions. The replication cohort adjusted for age, sex, the top two PCs, cell heterogeneity proportions, and batch effect. Statistical significance was calculated at $p \leq 1.06e-07$ (= 0.05/470871 total number of loci) for the discovery cohort. Statistical power ranged from 0.5847 to 0.7466 based on analysis phenotype [18]. A significance threshold of $p \leq 7.14e-03$ (= 0.05/7) was determined for the look-up analysis in the replication cohort.

Weighted Gene Co-expression Network Analysis (WGCNA) [19] was used to identify networks or modules of highly correlated DNA methylation loci and blood biomarkers or clinical traits (n = 27 traits; S1 Table). Modules were calculated using the blockwise module function in the WGCNA R package. Outlier samples were identified and removed to ensure reliable network construction. An appropriate soft threshold power was calculated for each stratum. A soft thresholding power of 20 in the AFR cohort and 26 in the EUR cohort was determined at a threshold $\geq 0.8$. These parameters were used in a signed-hybrid network model with the minimum number of loci set to 30 and a maximum block size set to 10,000. The loci comprising each module were represented by a weighted average, which is indicative of the first principle component in the analysis.

Statistical significance was calculated as $p \leq 3.93e-09$ (= 0.05/(470871 loci*27 traits)), while the suggestive threshold was set at $p < 1.00e-03$.

Gene ontology (GO) term enrichment was performed using GOrilla (Gene Ontology enRIchment anaLysis and visuaLizAtion) [20, 21] for the seven significant loci identified in the survival analysis with $p \leq 1.06e-07$. Statistical significance for GO term enrichment included terms with a false discovery rate (FDR) q-value $\leq 0.05$.

To further evaluate the biological mechanisms of statistically and marginally significant ($p < 1.00e-06$) CpG sites from the AFR VISP recurrent stroke and composite vascular event analyses, the Functional Mapping and Annotation of Genome-Wide Association Studies (FUMA GWAS) version 1.3.6a [22] online platform and the GENE2FUNC process was utilized. The EUR composite vascular event analysis resulted in two CpGs that met or exceeded the suggestive threshold and therefore, functional mapping was not performed. A total of 36 (AFR VISP recurrent stroke) and 30 (AFR VISP composite vascular event) gene names were combined, resulting in 57 unique gene names uploaded into FUMA for evaluation of gene

expression and enrichment of differentially expressed gene sets in tissues from the Genotype-Tissue Expression (GTEx) 8 RNA sequencing data [23]. Upon filtering by ENSEMBL identifiers, 53 genes were used in the GTEx analyses. A multiple testing correction was performed using a Benjamini-Hochberg adjustment. Statistical significance was calculated using a p-threshold of $p < 0.05$.

## Results

The baseline characteristics of 76 AFR and 104 EUR VISP study participants with and without recurrent stroke were compared using $\chi^2$ and t-tests for categorical and continuous variables, respectively, and are presented in Table 1. Of these individuals, 28 AFR and 32 EUR participants had an incident VISP recurrent stroke. In the AFR individuals, the average baseline age of those experiencing a recurrent stroke was five years older compared to those not having a recurrent stroke (65 years versus 60 years; p = 0.047). Of particular interest, those individuals having a VISP recurrent stroke had a more severe enrollment stroke compared to the non-recurrent control group, as measured on the modified Rankin stroke scale (RSS). While the enrollment criteria for VISP included mild-to-moderate RSS scores of 0–3, 35.7% of those suffering a recurrent stroke during VISP follow-up experienced an enrollment stroke with a RSS of 3, which is indicative of moderate disability, while 10.4% of the non-recurrent individuals had a similar enrollment stroke severity. Although not as extreme, VISP EUR participants had similar baseline age differences (70 years versus 68 years for VISP recurrence and nonrecurrence, respectively), as well as enrollment stroke severity compared with AFR participants. Approximately 44% of EUR VISP participants who experienced a recurrent stroke had RSS of 0 or 1, compared to nearly 71% of participants in the non-recurrent group having a similar score indicative of no significant disability (Table 1).

Ethnicity-stratified Cox PH analyses identified a total of seven methylation loci associated with time to event for recurrent stroke or composite vascular event (Tables 2 and 3). Four statistically significant loci were identified for days to VISP recurrent stroke. The most significant association was observed for cg04059318 (HR [95% CI] = 7.19 [3.55–14.57]; p = 4.52e-08), located on chromosome 10. Three additional loci, ch.2.81927627R (HR = 2.72 [1.89–3.93]; p = 9.11e-08), cg03584380 (HR = 5.41 [2.91–10.06]; p = 9.66e-08), and cg24875416 (HR = 2.43 [1.75–3.37]; p = 9.82e-08) were also implicated in the AFR recurrent stroke analysis (Table 2; Fig 1). There were no statistically significant loci in the days to VISP stroke recurrence analysis in EUR. cg03584380, an intronic locus of *ZDHHC6*, was the only locus to remain significant in the replication cohort from Vall d' Hebron University in Barcelona (HR = 1.83(1.21–2.77); p = 4.00e-03; Table 2). The ENCODE annotation and gene position for cg03584380 is presented in Fig 2. For the time to composite vascular event analysis, cg00076998 (HR = 5.58 [2.98–10.44]; p = 7.87e-08) and cg16758041 (HR = 3.44 [2.18–5.43]; p = 1.04e-07) were identified in AFR (Fig 3; Table 3), while cg02365967 (HR: 0.42 [0.31–0.58]; p = 8.08e-08) was identified in the EUR stratum (Fig 4; Table 3). Using Schoenfeld residual tests, we did not observe any evidence of violation to the proportional hazards assumptions for any of the base models indicated by a global $p < 0.05$ (p = 0.084 for AFR VISP recurrent stroke, p = 0.059 for AFR composite vascular endpoint, p = 0.320 for EUR VISP recurrent stroke, and p = 0.121 for EUR composite vascular endpoint analyses).

GO analysis was performed on the eight genes located closest to the significant loci identified in the survival analyses (*PTEN*, *KLLN*, *PIK3CB*, *HERC2*, *CTNNA2*, *ZDHHC6*, *STRIP1*, and *NDUFB6*). Nine terms describing biological processes and molecular function met a modest significance threshold ($p \leq 1.00e-03$), including prepulse inhibition (GO: 0060134; p = 8.22e-06), brain morphogenesis (GO: 0048854, p = 2.87e-05), and axonogenesis (GO:

**Table 1. Baseline demographics for VISP participants.**

| | AFR | | | EUR | | |
|---|---|---|---|---|---|---|
| | VISP Recurrent Stroke Cases | Nonrecurrent Stroke Controls | p[a] | VISP Recurrent Stroke Cases | Nonrecurrent Stroke Controls | p[a] |
| N | 28 | 48 | | 32 | 72 | |
| Age, yrs[b] | 65.18 (11.54) | 60.40 (8.94) | 0.047 | 70.03 (11.48) | 68.01 (10.81) | 0.391 |
| Sex | | | | | | |
| Male (%) | 17 (60.7) | 29 (60.4) | 1.000 | 18 (56.2) | 39 (54.2) | 1.000 |
| Female (%) | 11 (39.3) | 19 (39.6) | | 14 (43.8) | 33 (45.8) | |
| Treatment Arm | | | | | | |
| High-dose (%) | 14 (50.0) | 22 (45.8) | 0.910 | 14 (43.8) | 40 (55.6) | 0.368 |
| Low-dose (%) | 14 (50.0) | 26 (54.2) | | 18 (56.2) | 32 (44.4) | |
| Current Smoker (%) | 8 (28.6) | 13 (27.1) | 1.000 | 3 (9.4) | 9 (12.5) | 0.898 |
| Body Mass Index, kg/m$^2$ | 29.01 (5.45) | 30.35 (6.65) | 0.370 | 29.64 (7.21) | 28.52 (5.68) | 0.403 |
| Diabetes mellitus (%) | 9 (32.1) | 20 (41.7) | 0.562 | 13 (40.6) | 21 (29.2) | 0.356 |
| Myocardial infarction (%) | 3 (10.7) | 1 (2.1) | 0.274 | 0 (0.0) | 4 (5.6) | 0.419 |
| Recurrent stroke ever (%) | 28 (100.0) | 16 (33.3) | <0.001 | 32 (100.0) | 15 (20.8) | <0.001 |
| PNS | | | | | | |
| 0 | 12 (44.4) | 32 (66.7) | 0.139 | 19 (61.3) | 57 (79.2) | 0.212 |
| 1 | 8 (29.6) | 13 (27.1) | | 6 (19.4) | 10 (13.9) | |
| 2 | 4 (14.8) | 2 (4.2) | | 5 (16.1) | 4 (5.6) | |
| 3 | 2 (7.4) | 1 (2.1) | | 1 (3.2) | 1 (1.4) | |
| 4 | 1 (3.7) | 0 (0.0) | | 0 (0.0) | 0 (0.0) | |
| RSS | | | | | | |
| 0 | 2 (7.1) | 7 (14.6) | 0.001 | 4 (12.5) | 17 (23.6) | 0.071 |
| 1 | 16 (57.1) | 20 (41.7) | | 10 (31.2) | 34 (47.2) | |
| 2 | 0 (0.0) | 16 (33.3) | | 14 (43.8) | 16 (22.2) | |
| 3 | 10 (35.7) | 5 (10.4) | | 4 (12.5) | 5 (6.9) | |
| Hypertension (%) | 24 (85.7) | 42 (87.5) | 1.000 | 28 (87.5) | 56 (77.8) | 0.373 |
| Systolic Blood Pressure, mmHg | 144.34 (20.43) | 143.65 (20.16) | 0.886 | 146.77 (19.29) | 140.11 (19.67) | 0.112 |
| Diastolic Blood Pressure, mmHg | 80.09 (9.89) | 82.05 (10.12) | 0.414 | 81.34 (8.67) | 78.14 (9.81) | 0.115 |
| Creatinine, mg/dL | 1.16 (0.39) | 1.41 (0.68) | 0.076 | 1.26 (0.55) | 1.11 (1.00) | 0.44 |
| Total cholesterol, mg/dL | 190.09 (53.10) | 212.85 (55.33) | 0.107 | 206.48 (37.67) | 201.93 (48.04) | 0.641 |
| High-density lipoprotein, mg/dL | 42.17 (10.76) | 51.76 (16.99) | 0.016 | 45.17 (13.78) | 46.59 (16.10) | 0.676 |
| Triglycerides, mg/dL | 146.95 (105.06) | 162.46 (138.70) | 0.65 | 193.83 (110.43) | 201.89 (193.05) | 0.834 |
| Total plasma homocysteine, μmol/L | 15.20 (5.16) | 14.95 (6.62) | 0.862 | 15.58 (6.85) | 16.82 (12.82) | 0.607 |
| B6, nmol/L | 36.08 (33.36) | 22.33 (20.91) | 0.042 | 29.99 (19.26) | 43.03 (50.34) | 0.18 |
| B12, pmol/L | 421.63 (226.23) | 418.30 (190.21) | 0.947 | 354.97 (132.03) | 313.50 (124.92) | 0.144 |
| Folate, ng/mL | 31.92 (32.59) | 21.85 (9.26) | 0.057 | 20.84 (10.59) | 24.65 (11.25) | 0.122 |
| C-reactive protein, mg/L | 12.49 (8.92) | 12.64 (10.26) | 0.953 | 13.50 (8.41) | 11.37 (8.92) | 0.277 |
| Prothrombin fragments 1+2, nmol/L | 0.99 (0.56) | 1.25 (1.42) | 0.386 | 1.14 (0.63) | 1.14 (1.28) | 0.996 |
| Thrombin-antithrombin complex, μg/L | 9.11 (12.52) | 5.26 (3.49) | 0.099 | 6.47 (8.94) | 6.42 (6.73) | 0.98 |
| Thrombomodulin, ng/mL | 10.33 (18.47) | 11.74 (25.00) | 0.84 | 10.66 (22.63) | 8.34 (12.82) | 0.588 |
| Tissue plasminogen activator, ng/mL | 15.42 (10.61) | 15.78 (19.58) | 0.933 | 14.32 (15.14) | 18.07 (21.12) | 0.419 |

(*Continued*)

**Table 1.** (Continued)

| | AFR | | | EUR | | |
|---|---|---|---|---|---|---|
| | VISP Recurrent Stroke Cases | Nonrecurrent Stroke Controls | p[a] | VISP Recurrent Stroke Cases | Nonrecurrent Stroke Controls | p[a] |
| **von Willebrand Factor,** IU/L | 1551.15 (747.11) | 1222.32 (616.71) | 0.053 | 1284.78 (652.09) | 1315.66 (874.32) | 0.871 |

[a]p-value calculated using t-tests and $\chi^2$ for continuous and categorical traits, respectively.

[b]Continuous traits described as mean (SD). Categorical traits described as N (%).

**Abbreviations**: AFR- African descent stratum; EUR- European descent stratum; PNS- number of strokes prior to VISP enrollment; RSS- modified Rankin stroke scale.

0007409, p = 6.94e-04) (S2 Table). To further elucidate any biological implications of our findings, we performed functional annotation and mapping of the 57 unique genes with suggestive (p<1.00E-06) CpG loci from the Cox PH results on AFR VISP recurrent stroke and composite vascular endpoint analyses. Tissue analysis on 53 specific types from the GTEx project, revealed statistically significant differential down-regulated expression in the pancreas ($p_{adj}$ = 3.05e-08), putamen basal ganglia ($p_{adj}$ = 2.42e-06), left ventricle of the heart ($p_{adj}$ = 1.54e-05), liver ($p_{adj}$ = 8.75e-05), amygdala ($p_{adj}$ = 1.50e-04), caudate basal ganglia ($p_{adj}$ = 2.30e-04), hippocampus ($p_{adj}$ = 3.52e-04), nucleus accumbens basal ganglia ($p_{adj}$ = 9.30e-04), anterior cingulate cortex BA24 ($p_{adj}$ = 5.93e-04), substantia nigra ($p_{adj}$ = 6.02e-03), skeletal muscle ($p_{adj}$ = 2.72e-02), whole blood ($p_{adj}$ = 2.91e-02), hypothalamus ($p_{adj}$ = 2.93e-02), and the cortex ($p_{adj}$ = 3.89e-02) (Fig 5, Table 4), when compared to the background gene set from GTEx v8.

To evaluate comprehensive networks of methylation loci associated with stroke related clinical traits, WGCNA was performed in the two ethnic strata upon outlier removal, which resulted in 100 EUR and 73 AFR participants included. Twenty-seven traits, including baseline biomarker levels, stroke risk factors, and outcome statuses were included in the analyses (Table 1; S1 Table). In the AFR stratum, 106 modules were produced, of which six were observed in significant module-clinical trait associations (p≤3.93e-09; Table 5). The significant associations observed were between modules and the previous number of strokes experienced before VISP enrollment (r = -0.90; 5.00e-27), prothrombin fragments 1 + 2 (r = -0.79; p = 8.00e-17), thrombomodulin (r = -0.77; 1.00e-15), thrombin-antithrombin complex (r = -0.73; p = 3.00e-13), triglyceride levels (r = -0.71; p = 2.00e-12), and tissue plasminogen activator (r = -0.68; p = 3.00e-11). Additional module associations in AFR were detected with traits and outcomes including MI, hypertension, thrombomodulin, and thrombin-antithrombin complex. In the EUR analysis, 27 modules were produced; however, no module-trait associations observed met the significance threshold. Notable nominal associations in this stratum included modules correlated with composite vascular endpoint, VISP recurrent stroke, and

**Table 2. Significant methylation loci associated with time to VISP recurrent stroke in AFR (p≤1.06e-07).**

| | [a]Discovery Cohort: VISP | | | | | | Replication |
|---|---|---|---|---|---|---|---|
| Locus | CHR:BP | Locus Location | Gene | Mean Beta Value | HR (95% CI) | P | P |
| cg04059318 | 10:89622526 | TSS1500; Exon 1 | *PTEN; KLLN* | 0.07 | 7.19 (3.55–14.57) | **4.52E-08** | 0.665 |
| ch.2.81927627R | 2:82074116 | Downstream | *CTNNA2* | 0.09 | 2.72 (1.89–3.93) | **9.11E-08** | 0.178 |
| cg03584380 | 10:114206433 | Intron 1 | *ZDHHC6* | 0.06 | 5.41 (2.91–10.06) | **9.66E-08** | **0.004** |
| cg24875416 | 1:110577936 | Intron 1 | *STRIP1* | 0.08 | 2.43 (1.75–3.37) | **9.82E-08** | 0.784 |

[a]Discovery model adjusts for age, sex, top 10 genetic principal components, treatment arm, batch effect, and estimated cellular proportions.

**Abbreviations**: CHR- chromosome; BP-base position (hg19); TSS1500- covers from 200 to 1500 nucleotides upstream of transcriptional start site (TSS); HR-hazard ratio; CI- confidence interval.

**Table 3. Significant methylation loci associated with time to VISP composite vascular endpoint (p≤1.06e-07).**

| | | | [a]Discovery Cohort: VISP | | | | | Replication |
|---|---|---|---|---|---|---|---|---|
| Strata | Locus | CHR:BP | Locus Location | Gene | Mean Beta Value | HR (95% CI) | P | P |
| AFR | cg00076998 | 3:138553170 | Upstream | *PIK3CB* | 0.06 | 5.58 (2.98–10.44) | **7.87E-08** | 0.608 |
| | cg16758041 | 9:32573371 | TSS200 | *NDUFB6* | 0.07 | 3.44 (2.18–5.43) | **1.04E-07** | 0.738 |
| EUR | cg02365967 | 15:28473380 | Exon 35 | *HERC2* | 0.94 | 0.42 (0.31–0.58) | **8.08E-08** | 0.163 |

[a]Discovery model adjusts for age, sex, genetic principal components (first 4 in EUR; first 10 in AFR), treatment arm, batch effect, and estimated cellular proportions.
**Abbreviations**: CHR- chromosome; BP-base position (hg19); TSS200- region from the transcriptional start site (TSS) to 200 nucleotides upstream of TSS; HR-hazard ratio; CI- confidence interval.

total plasma homocysteine. Only two loci identified in the Cox PH models were included in any of the modules. While identified in the AFR survival analyses, cg04059318 and cg24875416 were included in the turquoise module for EUR. Interestingly, this was one of two modules nominally associated with VISP recurrent stroke or composite vascular endpoint (r = 0.43, p = 8.00e-06 and r = 0.44, p = 5.00e-06, respectively) (Table 5).

## Discussion

To the best of our knowledge, this study represents the first epigenome-wide association study evaluating association between time to recurrent stroke (or composite vascular event) and the degree of methylation. This study was performed in a subset of the VISP clinical trial

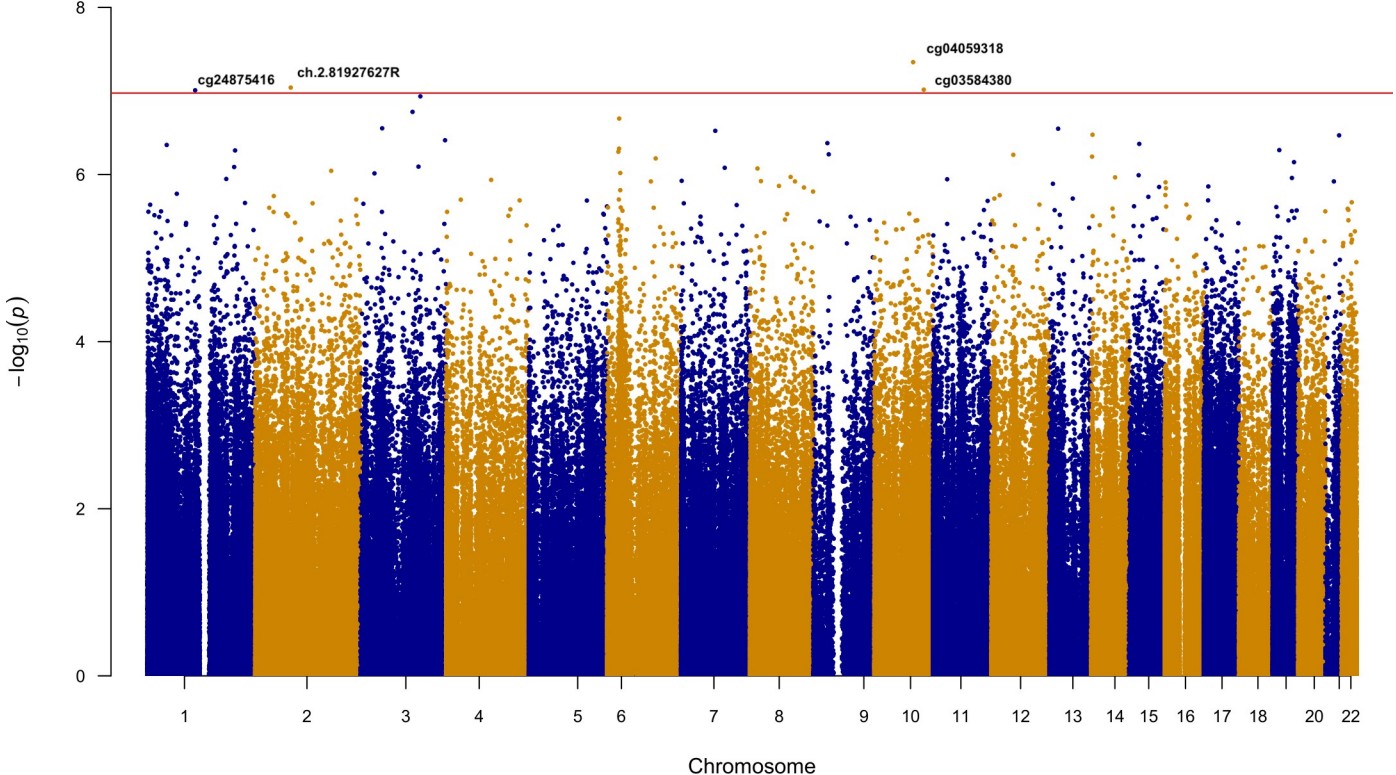

**Fig 1. Epigenome-wide Manhattan plot for time (days) to VISP recurrent stroke survival analysis in AFR.** Each point corresponds to the -log₁₀(P-value) for a CpG site at its specific chromosome location (y-axis). Horizontal line is indicative of epigenome-wide significance threshold (p≤ 1.06e-07).

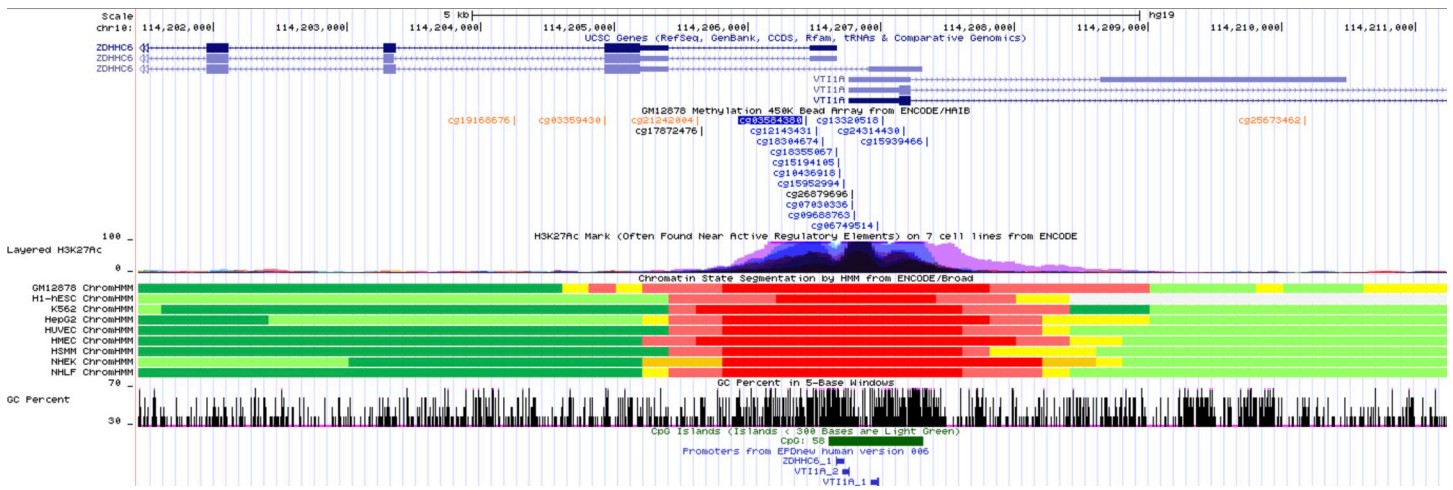

**Fig 2. ENCODE annotation of cg03584380 on intron 1 of *ZDHHC6*.** The annotation for cg03584380 includes CpG islands, cell line chromatin state (ChromHMM), H3K27Ac marks, and cell line methylation at CpG sites on the Methyl450 Bead Arrays from ENCODE/Hudson Alpha Institute for Biotechnology (ENCODE/HAIB; bright blue, purple, and orange CpGs correspond to unmethylation, partially methylated, and methylated states, respectively).

participants, providing a diverse cohort of individuals of both AFR and EUR descent. The inclusion of individuals of AFR descent is a strength of this study, since this population is 60% more likely to experience a recurrent stroke within two years compared to individuals of EUR descent, albeit likely mediated by stroke risk factors and comorbidities [24]. Seven loci reached

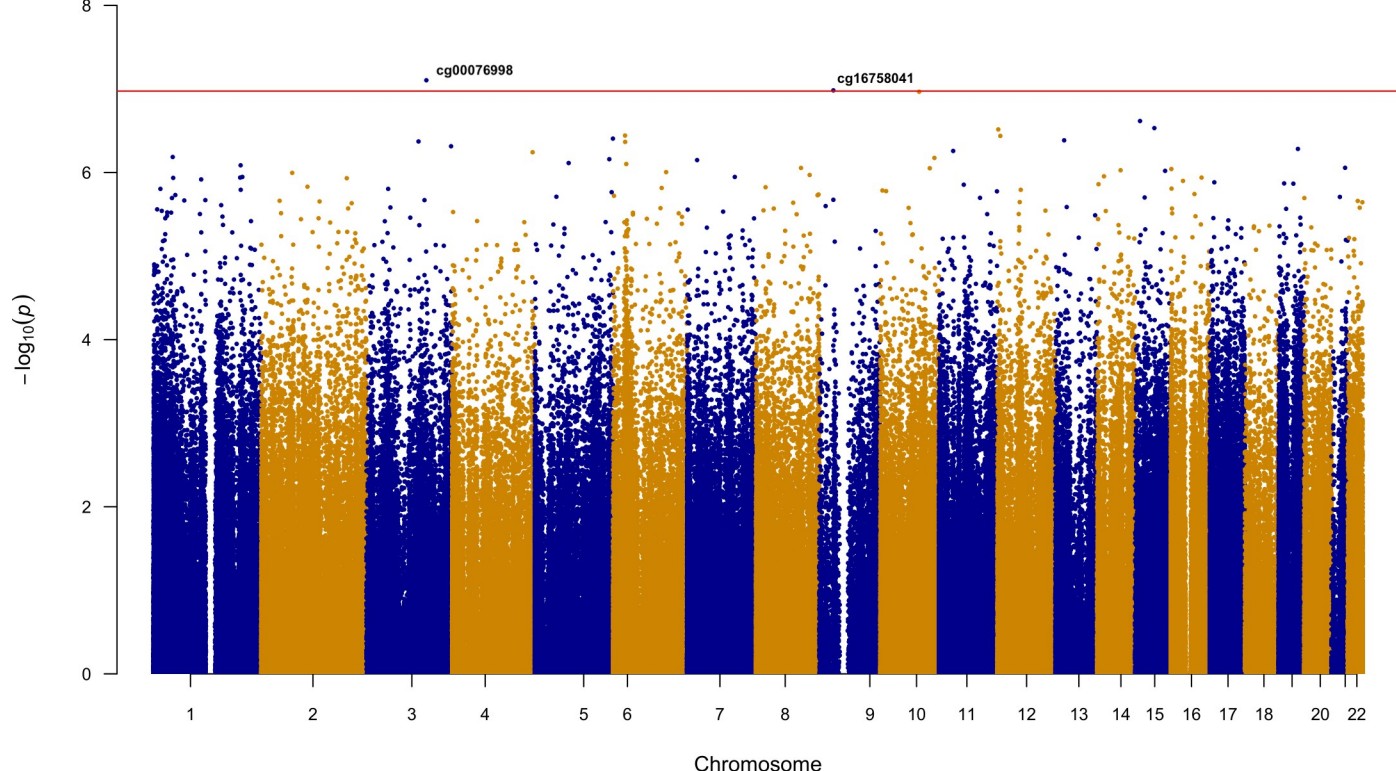

**Fig 3. Epigenome-wide Manhattan plot for AFR time to composite vascular event.** Horizontal line is indicative of epigenome-wide significance threshold (p≤ 1.06e-07).

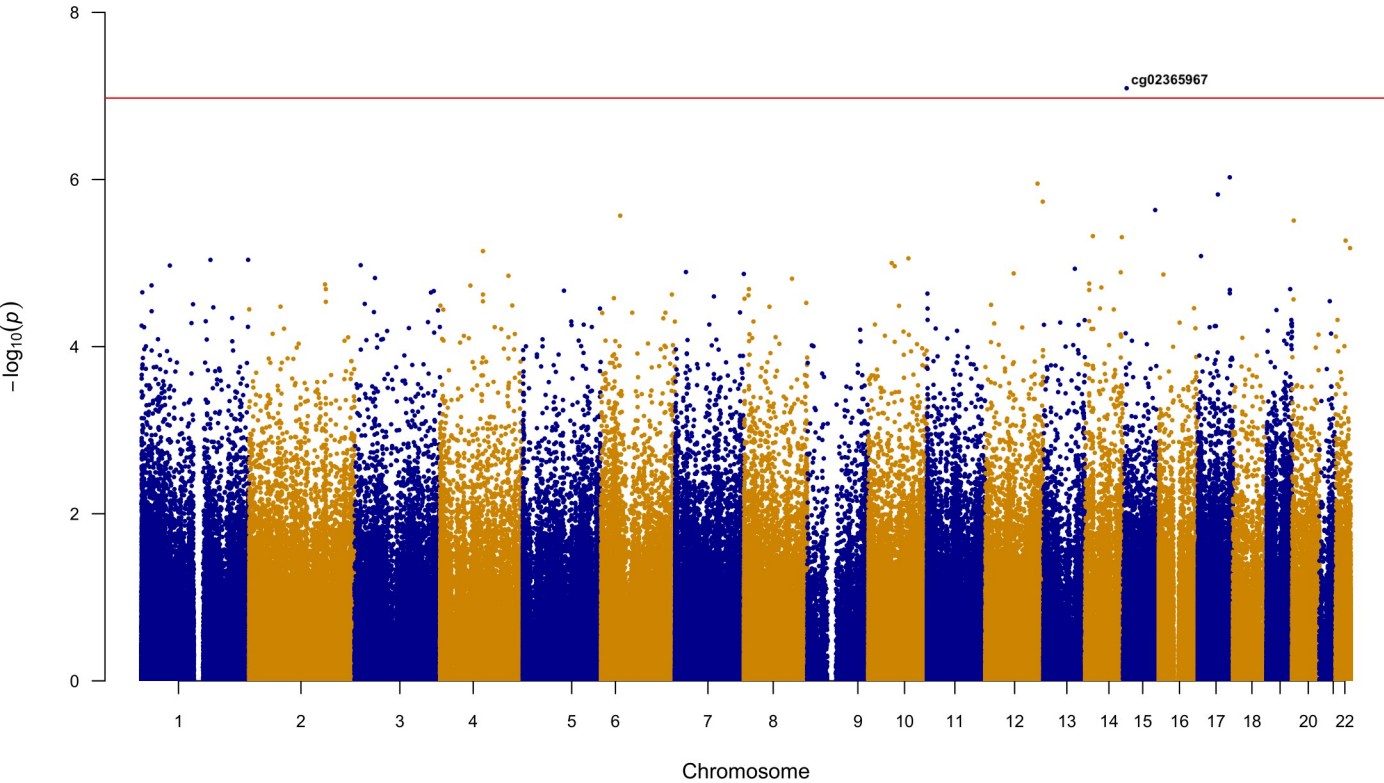

**Fig 4. Epigenome-wide Manhattan plot for EUR time to composite vascular event.** Horizontal line is indicative of epigenome-wide significance threshold (p≤ 1.06e-07).

or exceeded a Bonferroni corrected significance threshold, with cg03584380 statistically significant in AFR stroke recurrence and validated upon replication in the Vall d'Hebron University Hospital cohort. While similar trends were observed across phenotypes due to the strong correlation between the primary (VISP recurrent stroke) and secondary (composite vascular) outcomes of interest at these loci, results were not significant across separate cohorts, suggesting potential ethnic disparities, as demonstrated by the significant association between

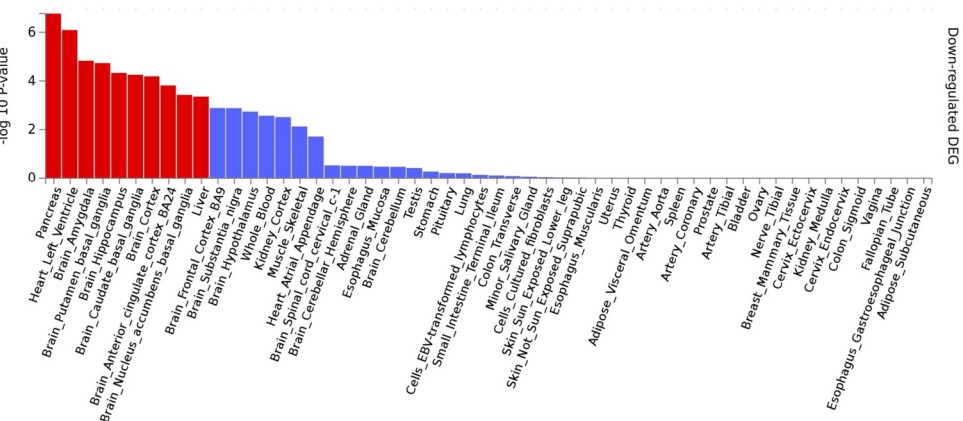

**Fig 5. Differentially expressed down-regulated genes from FUMA.** Red bars are indicative of statistically significant differentially expressed tissues based on Benjamini-Hochberg adjustment. Significance threshold p_adj<0.05.

**Table 4. Differentially expressed genes from FUMA using GTEx version 8 tissue expression.**

| General Tissue Type | Specific Tissue Type from GTEx | N genes[a] | N overlap[b] | p[c] | p_adj[d] |
|---|---|---|---|---|---|
| Pancreas | Pancreas | 9668 | 37 | 5.64E-10 | 3.05E-08 |
| Brain | Putamen basal ganglia | 7853 | 31 | 4.49E-08 | 2.42E-06 |
| Heart | Left ventricle | 9531 | 33 | 2.85E-07 | 1.54E-05 |
| Liver | Liver | 8059 | 29 | 1.62E-06 | 8.75E-05 |
| Brain | Amygdala | 7751 | 28 | 2.77E-06 | 1.50E-04 |
| Brain | Caudate basal ganglia | 6927 | 26 | 4.26E-06 | 2.30E-04 |
| Brain | Hippocampus | 7571 | 27 | 6.53E-06 | 3.52E-04 |
| Brain | Nucleus accumbens basal ganglia | 6479 | 24 | 1.72E-05 | 9.30E-04 |
| Brain | Anterior cingulate cortex BA24 | 6725 | 23 | 1.10E-04 | 5.93E-03 |
| Brain | Substantia nigra | 7223 | 24 | 1.11E-04 | 6.02E-03 |
| Muscle | Skeletal | 6908 | 22 | 5.03E-04 | 2.72E-02 |
| Blood | Whole blood | 6940 | 22 | 5.38E-04 | 2.91E-02 |
| Brain | Hypothalamus | 5967 | 20 | 5.43E-04 | 2.93E-02 |

[a]The number of background genes expressed in specific tissue from GTEx version 8.

[b]The number of overlap occurring between input genes and background.

[c]Unadjusted p-value from the hypergeometric test performed in FUMA.

[d]Benjamini-Hochberg adjusted p-value.

cg03584380 and stroke recurrence in the AFR and Spanish cohorts, but not in EUR (p = 4.07e-01). Within the AFR stratum, an increase in the beta values at the six significant loci were associated with shorter duration to recurrent stroke or composite vascular event, thus suggesting

**Table 5. Significant WGCNA module-trait associations.**

| Strata | Module | Trait | r | p[a] |
|---|---|---|---|---|
| **AFR** | **bisque4** | **Previous number of strokes** | **-0.90** | **5.00E-27** |
| **AFR** | **coral2** | **Prothrombin fragments 1 + 2** | **-0.79** | **8.00E-17** |
| **AFR** | **pink4** | **Thrombomodulin** | **-0.77** | **1.00E-15** |
| **AFR** | **coral** | **Thrombin-antithrombin complex** | **-0.73** | **3.00E-13** |
| **AFR** | **yellowgreen** | **Triglycerides** | **-0.71** | **2.00E-12** |
| **AFR** | **lightsteelblue** | **Tissue plasminogen activator** | **-0.68** | **3.00E-11** |
| AFR | darkturquiose | Myocardial infarction | 0.51 | 4.00E-06 |
| EUR | turquoise | Composite endpoint | 0.44 | 5.00E-06 |
| EUR | turquoise | VISP recurrent stroke | 0.43 | 8.00E-06 |
| AFR | honeydew1 | Myocardial infarction | 0.49 | 9.00E-06 |
| AFR | blue2 | Myocardial infarction | -0.49 | 1.00E-05 |
| AFR | coral1 | Myocardial infarction | -0.48 | 2.00E-05 |
| AFR | lightcyan1 | Myocardial infarction | -0.48 | 2.00E-05 |
| AFR | lightslateblue | Myocardial infarction | 0.48 | 2.00E-05 |
| AFR | coral | Thrombomodulin | -0.43 | 1.00E-04 |
| AFR | mediumpurple2 | Thrombin-antithrombin complex | -0.44 | 1.00E-04 |
| AFR | royalblue | Myocardial infarction | -0.42 | 3.00E-04 |
| EUR | cyan | Composite endpoint | 0.34 | 5.00E-04 |
| AFR | thistle3 | Hypertension | 0.39 | 7.00E-04 |
| EUR | cyan | Total plasma homocysteine | 0.33 | 8.00E-04 |

[a]Statisitical significance (**bold**): p≤ 3.93e-09.

that increased methylation at these loci could be indicative of earlier event recurrence. Three of these loci have implications in the phosphatidylinositol pathway, while the remaining three loci were located in genes or gene families linked to cardiovascular traits including the pathogenesis of cardiac disease and diabetes [25], regulation of hyperlipidemia and arteriosclerosis [26], and insulin sensitivity [27].

cg03584380 was the only locus replicated in our analyses. This methylation site is located in the first intron of *ZDHHC6*, the gene which encodes palmitoyltransferase ZDHHC6 [28] and mediates the palmitoylation of critical endoplasmic reticulum (ER) proteins, including calnexin and the inositol 1,4,5-trisphosphate receptor (ITPR1) [29]. Calnexin is an ER chaperone that has been implicated in cardiomyocyte viability and in ER stress, a prominent clinical feature of cardiovascular disease [29], while ITPR1 mediates the influx and release of intracellular $Ca^{2+}$ and is regulated by the palmitoylation cascade of ZDHHC16/ZDHHC6 [30, 31]. Additionally, ITPR1 can be phosphorylated by Akt kinase and further regulated by phosphatidylinositol 3-kinase (PI3K) [32]. Results from the ENCODE annotation suggest this region around cg03584380 is an active regulatory site. There are several active promoters for both the *ZDHHC6* and *VTI1A* genes as indicated by the red bars in Fig 2 in the chromatin state segmentation by the Hidden Markov Model (HMM) track, as well as a CpG island just downstream of this locus. Furthermore, there is evidence of increased H3K27 (lysine 27 of the H3 histone) acetylation, which also indicates enhanced transcription. Therefore, it is plausible that cg03584380 could regulate chromatin and histone states related to transcription and transcription factor binding, although future gene expression and functional work is needed.

Although the association between cg04059318 and AFR stroke recurrence was not validated in the EUR only analysis (p = 7.34e-02) or look-up efforts (p = 6.65e-01), it was the most statistically significant association detected in the VISP survival analysis. cg04059318 is located within the 5' untranslated region of *PTEN* and within an intron of *KLLN*. *KLLN* encodes KILLIN, a DNA-binding protein that inhibits DNA synthesis and mediates p53/TP53-induced apoptosis [28]. *PTEN* encodes PTEN, a tumor suppressor that regulates angiogenesis [33] and has been associated with a number of stroke-related clinical traits including triglycerides [34] and type 2 diabetes [35]. PTEN negatively regulates the Akt signaling pathway through intracellular phosphatidylinositol 3-phosphate (PI3P), which is of interest due to the neuroprotective properties of Akt against ischemia-induced damage [36, 37]. A third locus involved in the phosphatidylinositol pathway was identified in the composite vascular event analysis. cg00076998 is located upstream of *PIK3CB*, which encodes the catalytic subunit of phosphatidylinositol 3-kinase, beta (PI3Kβ). PI3K activates cellular signaling cascades through the generation of phosphatidylinositol (3,4,5)-trisphosphate and recruitment of Akt and phosphoinositide-dependent kinase-1 [38]. PI3K is involved in platelet activation signaling triggered by G-protein coupled receptors [28] and is required for platelet-induced aggregation induced by thrombin and thromboxane A2 (TxA2) [39]. PI3K/Akt signaling has been observed in regulation of vascular tone, or the degree of vasoconstriction experienced by a blood vessel, in both vascular endothelium and smooth muscle cells [40]. Exposure to homocysteine, a well-documented risk factor for atherosclerosis and stroke [41, 42], in endothelial cells can form the intermediate S-nitroso-homocysteine, a vasodilator implicated in reducing vascular tone and altering arteriole calcium levels [40] Collectively, these associations have identified a novel association with *ZDHHC6* and implicate PI3K/Akt signaling in time to stroke and/or vascular event following stroke [43, 44]. These findings could in part, help explain the ethnic disparities in stroke severity and recovery seen in African American patients; however, further functional analyses are needed to confirm the biological significance of our results.

cg02365967 was the only locus identified in our EUR cohort (composite vascular event; AFR composite p = 2.54e-01), and was the sole locus that exhibited hypermethylated beta values (mean β = 0.9411). Increased beta values at this locus were associated with longer duration to event, which differed from the other identified loci. cg02365967 is located in exon 35 of *HERC2*, which encodes the ubiquitin-protein ligase HERC2. This protein regulates ubiquitin-dependent retention of repair proteins and is a binding protein with SIRT1, a NAD-dependent protein deacetylase that plays a vasoprotective role in endothelial cells [45]. Increased levels of SIRT1 have been described to promote endothelial angiogenesis, enhance vasodilation, and suppress vascular inflammation [45]. Overexpression of SIRT1 in murine endothelial cells prevented hypertension and adverse arterial remodeling; however, a knockdown of HERC2 abolished any beneficial effects of SIRT1 [45], suggesting a neuroprotective regulatory role of HERC2.

Network analyses using WGCNA, identified six statistically significant associations between modules and clinical traits in the AFR stratum, including a correlation between a module comprised of 58 methylation loci within 38 genes and the previous number of strokes prior to VISP enrollment variable. Of these genes, *A2ML1*, *AGGF1*, *CBS*, *ECE1*, *GABRB3*, *GALNT2*, *MRPS6/SLC5A3*, and *SPG7* had documented associations with hypertension, ischemic stroke and methionine metabolism, atherosclerosis, and early-onset MI [46–48]. GO term enrichment of the genes identified in survival analyses, resulted in terms associated with prepulse inhibition (GO: 0060134), brain morphogenesis (GO: 0048854), axonogenesis (GO: 0007409), and postsynaptic assembly, organization, and regulation (GO: 0098698, GO: 2000463, GO: 0098815, and GO: 0099084). Prepulse inhibition (PPI) describes the regulated transmission of sensory information, while disrupted PPI has been linked to neurological disorders including Tourette's syndrome and Schizophrenia [49]. Brain morphogenesis generates and organizes anatomical structures of the brain, which can be a crucial process after ischemia. Axonogenesis, or the *de novo* generation of a neuron's long process [50], requires the melatonin MT2 receptor [51]. Deficits in MT2 signaling have been observed in a number of neurological disorders, including Alzheimer's disease, suggesting that axonogenesis may be beneficial in regard to the outcome of these disorders [51]. The GO results, in addition to the differential gene expression analysis in GTEx tissues, provide evidence of altered cerebro- and cardiovascular regulation.

Recurrent stroke is a vastly understudied phenotype and its etiology is not well understood. The utilization of this phenotype is a strength of this study; however, one limitation is the lack of formally adjudicated ischemic stroke subtype. We can conclude that due to the inclusion and exclusion criteria of VISP, the enrollment strokes were most likely lacunar or small vessel infarctions. DNA methylation profiles were generated from whole blood samples. Although not optimal due to cellular heterogeneity, whole blood provides a valuable resource that is potentially available for replication studies and represents a minimally invasive source for potential biomarker testing. To overcome this limitation, cellular proportions were calculated *in silico*, and used as covariates in the Cox PH models. Although our analyses detected seven statistically significant loci and replicated cg03584380 in the Vall d'Hebron University Hospital cohort, one constraint of this study that could not be fully addressed was the limited statistical power due to the modest sample sizes of both the discovery cohort and replication cohort. Furthermore, previous studies have shown that global methylation patterns differ across ethnicities, as well as within ethnic subgroups. In a study of 573 individuals from diverse Latino ethnic sub-groups, genetic ancestry explained approximately 75% of the variation in methylation between the sub-groups [52]. Therefore, the use of cohorts from different races or ethnicities should allow for the identification of methylation sites with global implications (affecting most ethnicities similarly) but will limit our ability to identify race/ethnicity-specific

associations. In conclusion, findings from this study provide insight into the relationship between the degree of DNA methylation and the duration to recurrent stroke and vascular event following a stroke and lay the foundation for further studies investigating these outcomes in diverse populations.

## Supporting information

**S1 Table. List of blood biomarkers and clinical traits used in WGCNA.**
(DOCX)

**S2 Table. Gene ontology results from significant methylation loci.**
(DOCX)

## Acknowledgments

The authors would like to thank the individuals who participated in the VISP study. Co-author Michele M. Sale passed away before the submission of the final version of this manuscript. The corresponding author (KLK) accepts responsibility for the integrity and validity of the data collected and analyzed.

## Author Contributions

**Conceptualization:** Nicole M. Davis Armstrong, Stephen R. Williams, Michèle M. Sale, Bradford B. Worrall, Keith L. Keene.

**Data curation:** Wei-Min Chen, Fang-Chi Hsu, Michèle M. Sale.

**Formal analysis:** Nicole M. Davis Armstrong, Fang-Chi Hsu, Michael S. Brewer, Natalia Cullell, Keith L. Keene.

**Funding acquisition:** Michèle M. Sale, Bradford B. Worrall, Keith L. Keene.

**Methodology:** Nicole M. Davis Armstrong, Wei-Min Chen, Fang-Chi Hsu, Michael S. Brewer, Stephen R. Williams, Bradford B. Worrall, Keith L. Keene.

**Resources:** Michèle M. Sale, Bradford B. Worrall, Keith L. Keene.

**Supervision:** Keith L. Keene.

**Validation:** Natalia Cullell, Israel Fernández-Cadenas.

**Writing – original draft:** Nicole M. Davis Armstrong, Keith L. Keene.

**Writing – review & editing:** Nicole M. Davis Armstrong, Wei-Min Chen, Fang-Chi Hsu, Michael S. Brewer, Natalia Cullell, Israel Fernández-Cadenas, Stephen R. Williams, Bradford B. Worrall, Keith L. Keene.

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
