## [Decision Letter · Decision Letter 0]

21 Apr 2021

PONE-D-21-06413

DNA methylation analyses identify an intronic ZDHHC6 locus associated with time to recurrent stroke in the Vitamin Intervention for Stroke Prevention (VISP) clinical trial.

PLOS ONE

Dear Dr. Keene,

Thank you for submitting your manuscript to PLOS ONE. After careful consideration, we feel that it has merit but does not fully meet PLOS ONE’s publication criteria as it currently stands. Therefore, we invite you to submit a revised version of the manuscript that addresses the points raised during the review process.

We look forward to receiving your revised manuscript.

Kind regards,

Giulia Bivona

Academic Editor

PLOS ONE

Journal Requirements:

3. We note that you are reporting an analysis of a microarray, next-generation sequencing, or deep sequencing data set. PLOS requires that authors comply with field-specific standards for preparation, recording, and deposition of data in repositories appropriate to their field. Please upload these data to a stable, public repository (such as ArrayExpress, Gene Expression Omnibus (GEO), DNA Data Bank of Japan (DDBJ), NCBI GenBank, NCBI Sequence Read Archive, or EMBL Nucleotide Sequence Database (ENA)). In your revised cover letter, please provide the relevant accession numbers that may be used to access these data. For a full list of recommended repositories, see http://journals.plos.org/plosone/s/data-availability#loc-omics or http://journals.plos.org/plosone/s/data-availability#loc-sequencing.

Reviewers' comments:

Reviewer's Responses to Questions

**Comments to the Author**

1. Is the manuscript technically sound, and do the data support the conclusions?

Reviewer #1: Yes

Reviewer #2: Yes

Reviewer #3: Yes

2. Has the statistical analysis been performed appropriately and rigorously? 

Reviewer #1: I Don't Know

Reviewer #2: Yes

Reviewer #3: Yes

3. Have the authors made all data underlying the findings in their manuscript fully available?

Reviewer #1: No

Reviewer #2: Yes

Reviewer #3: Yes

4. Is the manuscript presented in an intelligible fashion and written in standard English?

Reviewer #1: Yes

Reviewer #2: Yes

Reviewer #3: Yes

5. Review Comments to the Author

Reviewer #1: Dear author(s)

first, I would like to congratulate you for the tremendous work that has been established in this paper, and I am terribly sorry for the loss of one of the authors.

there are very minor points which I would like to suggest:

1. RESULTS; line 211. it is written scompared, please correct.

2. REFERENCE: please edit the fonts style to be similar with the rest of text.

3. FIGURES: please change the resolution of the figures as they are blurry and not clear. please cite the figures in text as (Fig 1) ...etc.

4. STYLE: please justify the text within and remove any unnecessary spaces

5. TABLES: kindly edit the tables captions font size and style. (refer to the guidelines)

6. ETHICS: if you have any ethics code, please state it in the ETHICS section.

7. Supporting information captions should be added after the REFERENCES section. Add Supporting information heading.

Reviewer #2: Summary:

The article describes a novel approach to elucidate the impact of epigenetic markers in disease, more specifically in recurrent stroke. The arguments are well structured, the paper is clear and the conclusions are important to the field. However, the cohort is too small to reach relevant conclusions and the heterogenicity of blood samples may need a bigger cohort to achieve consensus in the field.

Minor points:

1.- The quality of the figures is too low. I would recommend to improve it prior publication.

2.- Descriptions in figures and tables are vague.

3.- The criteria for the loci chosen for analysis is missing.

4.- In conclusions, an hypothesis to explain the differences between cohorts is missing.

Major points:

1.- The paper describes seven significant loci identified by methylation analysis. However, a deep analysis for those specific regions is missing. I recommend the inclusion of extra figures/ supplementary data highlighting the epigenetic characteristics of those regions, including a map of the genes nearby, with annotations of , for example, SUZ12 and histone modification marks, to inquire into the relevance of the regions reported and the origin of the methylation differences.

Available datasets could help to generate extra figures to show histone modification marks, methylation profiles and GWAS enrichment in the regions reported. Also, to use a primary cell line of relevance to do a ChIP-seq of interesting marks could help to show the “normal” methylation profile and epigenetic characteristics that we would expect in healthy population without the bias of the blood heterogeneity and then we would be able to compare it to the reported cohort.

Reviewer #3: The MS is sound and I can not visualize any new experiment which could be necessary to strenght the conclusions.The paper is solid and the experiments were well thought.Even if it s a descriptive paper it has enough merits to be published

6. PLOS authors have the option to publish the peer review history of their article (what does this mean?). If published, this will include your full peer review and any attached files.

Reviewer #1: **Yes: **Dr Noora R. Al-Snan

Reviewer #2: No

Reviewer #3: No

---

## [Author Response · Author response to Decision Letter 0]

14 Jun 2021

The authors express their appreciation and gratitude to the editor and reviewers for their thoughtful critiques. Below, please find our itemized responses which correspond to bold underlined text and corresponding tracked changes throughout the manuscript. 

Review Comments to the Author

Reviewer #1: Dear author(s)

first, I would like to congratulate you for the tremendous work that has been established in this paper, and I am terribly sorry for the loss of one of the authors.

there are very minor points which I would like to suggest:

Response: Thank you for your acknowledgement regarding the tremendous work and your condolences.

1. RESULTS; line 211. it is written scompared, please correct.

Response: We have made the appropriate edit. 

2. REFERENCE: please edit the fonts style to be similar with the rest of text.

Response: We have made the appropriate edits to the references.

3. FIGURES: please change the resolution of the figures as they are blurry and not clear. please cite the figures in text as (Fig 1) ...etc.

Response: We improved the quality of the figures and uploaded the figures to the PLOS Preflight Analysis and Conversion Engine (PACE) digital diagnostic tool to ensure they meet the PLOS requirements. Additionally, we addressed the figure citation in the text accordingly. 

4. STYLE: please justify the text within and remove any unnecessary spaces

Response: We have formatted the text accordingly by justifying the text and removing unnecessary spaces.

5. TABLES: kindly edit the tables captions font size and style. (refer to the guidelines)

Response: We have edited the table and figure captions, titles, font, and style accordingly.

6. ETHICS: if you have any ethics code, please state it in the ETHICS section.

Response: No additional ethics codes, beyond our IRB statements have been included.

7. Supporting information captions should be added after the REFERENCES section. Add Supporting information heading.

Response: We have added the Supporting information heading and information after the REFERENCES section.

Reviewer #2: Summary:

The article describes a novel approach to elucidate the impact of epigenetic markers in disease, more specifically in recurrent stroke. The arguments are well structured, the paper is clear and the conclusions are important to the field. However, the cohort is too small to reach relevant conclusions and the heterogenicity of blood samples may need a bigger cohort to achieve consensus in the field.

Minor points:

1.- The quality of the figures is too low. I would recommend to improve it prior publication.

Response: We improved the quality of the figures and uploaded the figures to the PLOS Preflight Analysis and Conversion Engine (PACE) digital diagnostic tool to ensure they meet the PLOS requirements.

2.- Descriptions in figures and tables are vague.

Response: We have edited the table and figure captions, titles, font and style accordingly.

3.- The criteria for the loci chosen for analysis is missing.

Response: We have edited the text as follows (Methylation Data Generation section): 

“Illumina Infinium Human Methylation450k BeadChip microarrays were used to interrogate CpG sites across the genome and the resulting intensity files were analyzed using GenomeStudio. Beta scores were generated as the ratio of methylated intensities divided by the sum of the methylation and unmethylated intensities (i.e. the proportion of total signal from the methylation specific probe). Probes were removed if they failed to hybridize (detection p>5%) or if they were located on sex chromosomes. The filtered beta scores underwent stratified quantile normalization with the minfi package in R [11-13]. A total of 470,871 autosomal CpGs were used in subsequent analyses.” 

4.- In conclusions, an hypothesis to explain the differences between cohorts is missing.

Response: We have edited the text as follows (Discussion section): 

“Furthermore, previous studies have shown that global methylation patterns differ across ethnicities, as well as within ethnic subgroups. In a study of 573 individuals from diverse Latio ethnic sub-groups, genetic ancestry explained approximately 75% of the variation in methylation between the sub-groups [52]. Therefore, the use of cohorts from different race/ethnicities should allow for the identification of methylation sites with global implications (affecting most ethnicities similarly) but will limit our ability to identify race/ethnicity-specific associations.

Major points:

1.- The paper describes seven significant loci identified by methylation analysis. However, a deep analysis for those specific regions is missing. I recommend the inclusion of extra figures/ supplementary data highlighting the epigenetic characteristics of those regions, including a map of the genes nearby, with annotations of , for example, SUZ12 and histone modification marks, to inquire into the relevance of the regions reported and the origin of the methylation differences.

Available datasets could help to generate extra figures to show histone modification marks, methylation profiles and GWAS enrichment in the regions reported. Also, to use a primary cell line of relevance to do a ChIP-seq of interesting marks could help to show the “normal” methylation profile and epigenetic characteristics that we would expect in healthy population without the bias of the blood heterogeneity and then we would be able to compare it to the reported cohort.

Response: We have added additional in silico data to help further elucidate any biological implications of our findings. The ENCODE annotation and gene position for the most significant site, cg03584380 is presented in Fig 2. We performed functional annotation and mapping of the 57 unique genes with suggestive (p<1.00E-06) CpG loci from the Cox PH results on AFR VISP recurrent stroke and composite vascular endpoint analyses. Tissue analysis on 53 specific types from the GTEx project, revealed statistically significant differential down-regulated expression in the pancreas (padj=3.05e-08), putamen basal ganglia (padj=2.42e-06), left ventricle of the heart (padj=1.54e-05), liver (padj=8.75e-05), amygdala (padj=1.50e-04), caudate basal ganglia (padj=2.30e-04), hippocampus (padj=3.52e-04), nucleus accumbens basal ganglia (padj=9.30e-04), anterior cingulate cortex BA24 (padj=5.93e-04), substantia nigra (padj=6.02e-03), skeletal muscle (padj=2.72e-02), whole blood (padj=2.91e-02), hypothalamus (padj=2.93e-02), and the cortex (padj=3.89e-02) (Fig 5, Table 4).

Reviewer #3: The MS is sound and I can not visualize any new experiment which could be necessary to strenght the conclusions.The paper is solid and the experiments were well thought.Even if it s a descriptive paper it has enough merits to be published

---

## [Editor Report · Decision Letter 1]

30 Jun 2021

DNA methylation analyses identify an intronic ZDHHC6 locus associated with time to recurrent stroke in the Vitamin Intervention for Stroke Prevention (VISP) clinical trial.

PONE-D-21-06413R1

Dear Dr. Keith L. Keene,

We’re pleased to inform you that your manuscript has been judged scientifically suitable for publication and will be formally accepted for publication once it meets all outstanding technical requirements.

Kind regards,

Giulia Bivona

Academic Editor

PLOS ONE
---

## [Editor Report · Acceptance letter]

2 Jul 2021

PONE-D-21-06413R1 

DNA methylation analyses identify an intronic *ZDHHC6* locus associated with time to recurrent stroke in the Vitamin Intervention for Stroke Prevention (VISP) clinical trial. 

Dear Dr. Keene:

I'm pleased to inform you that your manuscript has been deemed suitable for publication in PLOS ONE. Congratulations! Your manuscript is now with our production department. 

Kind regards, 

on behalf of

Dr. Giulia Bivona 

Academic Editor

PLOS ONE